# Progerin Expression Induces Inflammation, Oxidative Stress and Senescence in Human Coronary Endothelial Cells

**DOI:** 10.3390/cells9051201

**Published:** 2020-05-12

**Authors:** Guillaume Bidault, Marie Garcia, Jacqueline Capeau, Romain Morichon, Corinne Vigouroux, Véronique Béréziat

**Affiliations:** 1Institut Hospitalo-Universitaire de Cardiométabolisme et Nutrition (ICAN), RHU CARMMA, Centre de Recherche Saint-Antoine, INSERM UMR_S 938, Sorbonne Université, 75012 Paris, France; guillaume.bidault84@gmail.com (G.B.); garcia.marie@live.fr (M.G.); jacqueline.capeau@inserm.fr (J.C.); corinne.vigouroux@inserm.fr (C.V.); 2Cytometry and Imagery platform Saint-Antoine (CISA), Inserm UMS30 Lumic, 75012 Paris, France; romain.morichon@sorbonne-universite.fr; 3Laboratoire Commun de Biologie et Génétique Moléculaires, Hôpital Saint-Antoine, Assistance Publique-Hôpitaux de Paris, 75012 Paris, France; 4Service d’Endocrinologie, Diabétologie et Endocrinologie de la Reproduction, Centre National de Référence des Pathologies Rares de l’Insulino-Sécrétion et de l’Insulino-Sensibilité (PRISIS), Hôpital Saint-Antoine, Assistance Publique-Hôpitaux de Paris, 75012 Paris, France

**Keywords:** Hutchinson–Gilford progeria syndrome, *LMNA*, progerin, lamin A, atherosclerosis, endothelial dysfunction, inflammation, prenylation, aging

## Abstract

Hutchinson–Gilford progeria syndrome (HGPS) is a rare premature aging disorder notably characterized by precocious and deadly atherosclerosis. Almost 90% of HGPS patients carry a LMNA p.G608G splice variant that leads to the expression of a permanently farnesylated abnormal form of prelamin-A, referred to as progerin. Endothelial dysfunction is a key determinant of atherosclerosis, notably during aging. Previous studies have shown that progerin accumulates in HGPS patients’ endothelial cells but also during vascular physiological aging. However, whether progerin expression in human endothelial cells can recapitulate features of endothelial dysfunction is currently unknown. Herein, we evaluated the direct impact of exogenously expressed progerin and wild-type lamin-A on human endothelial cell function and senescence. Our data demonstrate that progerin, but not wild-type lamin-A, overexpression induces endothelial cell dysfunction, characterized by increased inflammation and oxidative stress together with persistent DNA damage, increased cell cycle arrest protein expression and cellular senescence. Inhibition of progerin prenylation using a pravastatin–zoledronate combination partly prevents these defects. Our data suggest a direct proatherogenic role of progerin in human endothelial cells, which could contribute to HGPS-associated early atherosclerosis and also potentially be involved in physiological endothelial aging participating to age-related cardiometabolic diseases.

## 1. Introduction

Hutchinson–Gilford progeria syndrome (HGPS, OMIM #176670) is a premature aging syndrome due to de novo heterozygous substitutions in the *LMNA* gene. Within childhood, HGPS patients develop several features observed in the elderly population, notably a deadly premature atherosclerosis [1,2,3].

Alternative splicing of *LMNA* transcripts results in lamin A and C nuclear proteins, that are intermediate filaments that maintain nuclear architecture and regulate DNA replication and repair and gene expression [4]. Of relevance, while lamin C does not require posttranslational modifications, lamin A is synthesized as a precursor protein referred to as prelamin A. Prelamin A maturation requires the transient attachment of a lipid anchor, a farnesyl group, normally lost following the removal of the fifteen C-terminal amino acids of the protein by the metalloprotease ZMPSTE24 [5]. The most common mutation causing HGPS (*LMNA* c.1824 C > T) creates an aberrant splicing site resulting in a deletion of 50 amino acids, including the ZMPSTE24 cleavage site [1,2,6]. The truncated protein, named progerin, cannot be properly cleaved and retains its farnesyl anchor [7].

The pathophysiological mechanisms of atherosclerosis in HGPS remain elusive. Limited autopsy reports indicated that a dramatic loss of vascular smooth muscle cells (VSMCs) with fibrosis and advanced calcification of the vascular wall are common features of HGPS patients’ arteries [8,9]. These alterations were confirmed in HGPS mouse models, with large arteries showing a dramatic depletion of VSMCs and major extracellular matrix remodeling [10,11,12]. Given these observations, the majority of the research on atherosclerosis in HGPS focused on VSMC defects.

Endothelial cell dysfunction is considered as the initial step of atherosclerosis development, in keeping with the major importance of the endothelium in maintaining vascular homeostasis [13]. Previous studies reported that progerin accumulates in HGPS patients’ endothelial cells [9,14]. Recently, it has been reported that progerin alters endothelial cell function in mouse models in vivo, causing impaired mechanotransduction and a reduction of the atheroprotective endothelial nitric oxide synthase activity [15]. These alterations could participate in the severe contractile impairment observed in HGPS patients [16].

Endothelial cell inflammation and senescence have been shown to increase susceptibility to atherosclerosis during normal aging [17] and could be important contributing factors to insulin resistance and aging-related systemic metabolic dysfunctions [18]. Expression of progerin has been reported in atherosclerotic coronary arteries from aging individuals [9,19]. However, whether progerin expression in human endothelial cells can be involved in the senescence and proinflammatory features associated with vascular aging is currently unknown.

Therefore, the objective of this study is to evaluate the impact of progerin expression in human endothelial cells. We exogenously expressed progerin or wild-type (WT)-prelamin A in primary cultures of human coronary endothelial cells. Our data demonstrate that progerin but not WT-prelamin A overexpression in endothelial cells recapitulates some features of aging-associated endothelial cell dysfunction, including a proinflammatory phenotype and oxidative stress together with persistent DNA damage, increased cell cycle arrest protein expression and cellular senescence. In accordance with a pathogenic role for the persistence of the farnesyl moiety of progerin, pharmacological inhibition of farnesylation with the combination of an aminobisphosphonate and an HMG-CoA reductase (3-hydroxy-3-methyl-glutaryl-coenzyme A reductase) inhibitor (zoledronate and pravastatin, ZOPRA) partly restored endothelial cell function.

## 2. Materials and Methods

### 2.1. Cell Culture and Treatment

HCAECs (human coronary artery endothelial cells) and endothelial cell growth medium were purchased from Promocell (Heidelberg, Germany). The cells used in this study were issued from healthy nonobese adult donors [20]. HCAECs were seeded on 0.2%-gelatin-coated plastic dishes. When indicated, transduced cells were treated with the combination of pravastatin (1 µM) and zoledronate (1 µM) (Sigma Aldrich, St Louis, MO, USA). Vehicle-treated cells were used as controls.

### 2.2. Adenovirus Production and Adenoviral-Mediated Expression of WT-Prelamin A or Progerin in HCAECs

Δ50 prelamin A (progerin) cDNA was obtained from GeneArt (Thermo Fisher scientific, Invitrogen Corporation, San Diego, CA, USA) from full-length rat prelamin A cDNA. cDNA of WT-prelamin A or progerin was integrated in a pAd5 plasmid vector, under the control of the cytomegalovirus (CMV) promoter. HEK 293 cells were transfected with recombinant pAd5 in order to produce adenoviral particles containing encoding sequences of WT-prelamin A or progerin, or with pAd5 empty vectors. Then, the recombinant adenoviruses were amplified in HEK 293 cells and purified by two consecutive cesium chloride (CsCl) centrifugation steps. CsCl was removed by gel filtration through PD10 columns (Amersham, GE Healthcare Europe, Velizy-Villacoublay, France) and collected as previously described [20]. Virus stocks were stored at −80 °C in phosphate-buffered saline (PBS) containing 15% glycerol. The titer of virus stocks was determined by quantitative real-time polymerase chain reaction (RT-PCR).

Early-confluent HCAECs were transduced with a multiplicity of infection of one virus particle per cell with Flag-tagged recombinant adenovirus containing WT-prelamin A or progerin. Nontransduced cells (control) or cells transduced with empty adenoviral vector (AdNul) were used as controls. Transduction efficiency, assessed for each adenoviral construct by counting Flag-positive cells, was not significantly different in each experiment between WT- and progerin-transduced cells (*p* = 0.6424) (Appendix A). All the experiments were performed 72 h after cell transduction.

### 2.3. Progerin Detection and Nuclear Shape

Protein extracts were subjected to sodium dodecyl sulphate-polyacrylamide gel electrophoresis (SDS-PAGE) and Western blotting. Antibodies were directed against lamin-A/C (SC-7292; Santa Cruz Biotechnologies Inc., Santa Cruz, CA, USA), which also recognizes progerin, revealed as an additional band, and β-actin (A-5441; Sigma-Aldrich). Protein bands were visualized by enhanced chemiluminescence (Pierce, Thermo Fisher scientific).

For immunofluorescence studies, HCAECs grown on 0.2%-gelatin-coated (Sigma-Aldrich) coverslips were fixed in methanol for 10 min at −20 °C. Antibodies directed against Flag (F-1804, Sigma-Aldrich) and lamin-A-specific antibodies (SC-20680, Santa Cruz Biotechnologies Inc.) were revealed by using secondary antibodies coupled to Alexa fluor 568 or 488 (Invitrogen Corporation) and counterstained with Hoechst 33342 (Sigma-Aldrich). Cells were visualized by Leica (Wetzlar, Germany) TCS SP2 lign 142 microscope at 100× magnification. Images were acquired with Leica confocal software.

Misshapen nuclei (blebs, invaginations, abnormal size) were quantified using lamin A and DNA (Hoechst 33342) stained cells and expressed as percentage of total nuclei.

### 2.4. Endothelial Dysfunction and Inflammation

Total RNA was isolated from cultured cells using RNeasy kit (Qiagen, Valencia, CA, USA), according to manufacturer instructions. Reverse transcription of mRNA to cDNA was performed using 500 ng of mRNA following manufacturer instruction (Applied Biosystems, Thermo Fisher Scientific). mRNA expression was analyzed by real-time PCR as described previously [20]. The sequence of the primers used is presented in Table 1. Hypoxanthine-phospho-ribosyl-transferase expression (HPRT) was used as an internal standard for mRNA expression.

Interleukin-6 (IL6), -1β (IL1β), and -8 (IL8/CXCL8), monocyte chemoattractant protein-1 (MCP1/CCL2), and vascular cell and intercellular adhesion molecule-1 (VCAM1 and ICAM1) secretion were measured from 24 h HCAEC culture supernatant with multiplexed bead-based immunoassays (Procarta, Affimetrix, Santa Clara, CA, USA) on a Bio-plex 200 system (Bio-Rad laboratories Inc., Hercules, CA, USA), using Bio-Plex Manager 4.1 software. Results were normalized to the protein content of the well. The detection limit was 10 pg/mL for all markers. IL1β secretion was assessed by enzyme-linked immunosorbent assay (ELISA), according to manufacturer instructions (BioLegend, San Diego, CA, USA).

### 2.5. Peripheral Blood Mononuclear Cell (PBMC) Adhesion Assay

PBMC adhesion assay was performed as previously described [20]. Briefly, fresh human PBMCs obtained from four different healthy blood donors were isolated by density gradient centrifugation (Ficollplaque plus, 17-1440-02, GE Healthcare Europe). PBMCs were counted and labeled for 30 min with fluorescent tracer calcein acetoxymethyl ester (10 μmol L^−1^) (Sigma-Aldrich). Labeled PBMCs were added for 1 h at 37 °C to previously transduced HCAECs, cultured for 24 h in serum-free endothelial cell basal medium (Promocell). After washing with PBS, adherent PBMCs were counted in five different fields (Leica TCS SP microscope, 40× magnification). Results were normalized to HCAECs protein level for each condition.

### 2.6. Oxidative Stress Assay

The production of reactive oxygen species (ROS) was assayed by quantifying the oxidation of 5-6-chloromethyl-2,7-dichlorodihydro-fluorescein diacetate (CM-H2DCFDA, Invitrogen Corporation) on a plate fluorescence reader at 520–595 nm, normalized to the protein level, as previously described [20].

### 2.7. Cellular Senescence Assay

Nuclear foci secondary to DNA Double-Strand Breaks (DSBs) were visualized by immunofluorescence with antibodies directed against Ser139-phosphorylated histone variant H2A (γ-H2AX, 05-636, Merk, Sigma-Aldrich) in transduced HCAECs. Nuclear DNA was stained with di-amidino-2-phenylindole hydrochloride staining (DAPI, Sigma-Aldrich).

The blue staining produced by SA-β-galactosidase’s hydrolysis of 5-bromo-4-chloro-3-indolyl-β-d-galactopyranoside (X-Gal, Sigma-Aldrich) was used as a biomarker of cellular senescence [21]. To quantify SA-β-galactosidase activity, cells were incubated with appropriate buffer solution containing X-Gal at pH 6, as described previously [22]. The proportion of positive (blue) cells was determined. The protein expression of cell cycle arrest markers was detected by incubation with specific primary antibodies for p53 (ab1101, Abcam, Cambridge, UK) and p21 (#2947, Cell Signaling, Danvers, MA, USA), as previously described [20].

### 2.8. Statistical Analysis

All experiments were performed at least three times on HCAECs issued from three different donors. All results are expressed as means ± standard error of mean (SEM). Statistical significance was determined using an analysis of variance (ANOVA) test followed by a Dunnett’s multiple comparison test to determine differences with WT-prelamin-A overexpressing cells. For ZOPRA experiments, data was analyzed using a two-way ANOVA followed by a Tukey post hoc test to assess differences against WT-prelamin A overexpression or a Sidak post hoc test to determine ZOPRA effect for each condition. *p* < 0.05 was considered as significant. Statistical analyses were performed using the Graphpad Prism (San Diego, CA, USA) 8.4.1 software.

## 3. Results

### 3.1. Progerin Accumulates at the Nuclear Periphery and Disrupts the Nuclear Shape of Endothelial Cells

In order to determine the effect of progerin on human endothelial cells, we transiently transduced primary human coronary artery endothelial cells (HCAECs) with an adenoviral vector containing the cDNA of Flag-tagged WT-prelamin A or progerin. The transduction efficiency, determined by the number of Flag-positive cells, did not differ between WT- and progerin-overexpressing cells (Appendix A). We confirmed by Western blot the overexpression of exogenous WT-prelamin A and progerin in transfected endothelial cells (Figure 1A).

We then assessed the localization of exogenous lamin A and progerin by confocal microscopy. As expected, flagged lamin A properly localized in the nucleus, fully colocalized with endogenous lamin A and nuclear architecture appeared normal (Figure 1B). Progerin was also properly addressed to the nucleus but it accumulated specifically at the nuclear rim, causing nuclear blebs and invaginations (Figure 1B,C), as previously observed in other models [23].

### 3.2. Progerin Expression Induces Endothelial Cell Inflammation and Reduces NO Synthase Expression

Endothelial cell inflammation contributes to atherosclerosis [24]. In HCAECs, the overexpression of progerin, but not WT-prelamin-A, increased the expression (Appendix A) and secretion of the proinflammatory cytokines IL6 and IL1β and of the chemokines and adhesion molecules CXCL8, CCL2, ICAM1 and VCAM1 (Figure 2A). We previously observed that, in HCAECs, secreted and surface expressions of VCAM-1 and ICAM-1 were related together [25]. Therefore, it is likely that increased gene expression and secretion of VCAM-1 and ICAM-1 are also associated with an increased level of these adhesion molecules at the surface of cells expressing progerin.

We then assessed whether the proinflammatory and proadhesion state of progerin-overexpressing HCAECs could enhance adhesion to the endothelial cell layer of peripheral blood mononuclear cells (PBMCs), formed by the entire mononuclear fraction of white blood cells isolated from healthy donors. We observed that the adhesion of PBMCs to progerin-expressing HCAECs was higher than that to control HCAECs (Control, AdNul, WT) (Figure 2B).

Finally, we observed a reduced gene expression of endothelial NO synthase (*NOS3*), a hallmark of endothelial dysfunction [17], in HCAECs expressing progerin, as previously observed in murine endothelial cells expressing progerin [15].

All together, these results demonstrated that progerin overexpression induced endothelial dysfunction, characterized by increased inflammation and proadhesion features and reduced endothelial NO synthase expression.

### 3.3. Endothelial Expression of Progerin Induces Oxidative Stress, DNA Damage and Senescence

Endothelial oxidative stress, induced by the imbalance between production and buffering of reactive oxygen species (ROS), is an important contributor to atherogenesis, notably during aging [26,27], and is a feature of progerin-expressing cells [28]. In our model, progerin expression increased ROS level (Figure 3A), together with induction of the cellular stress marker DDIT3 (C/EBP homologous protein) (Figure 3B).

Oxidative stress has been shown to induce DNA damage and endothelial dysfunction [29]. HGPS patients’ cells are prone to DNA damage accumulation [29]. We therefore posited that progerin overexpression in endothelial cells will favor the accumulation of DNA damage. In line with our hypothesis, progerin-overexpressing HCAECs accumulated more DNA double-strand breaks (DSBs) than WT-prelamin A-overexpressing cells, as assessed by the staining of phosphorylated histone γ-H2AX (Figure 3C).

Accumulation of unrepaired DNA damage triggers an inhibition of cell cycle progression, eventually leading to cell senescence [30]. Accordingly, we observed an upregulation of the cell cycle arrest proteins p53 and p21 in progerin-expressing HCAEC (Figure 3D). Although expression of progerin tends to induce a decrease in the protein amount per dish (Appendix A), this mild effect is unlikely to outweigh the magnitude of the observed differences in the expression of cell cycle arrest protein, as well as in the secretion of cytokines and adhesion assays presented above.

Finally, we observed that the number of cells presenting senescence-activated β-galactosidase activity was enhanced in HCAECs expressing progerin but not in those expressing WT-prelamin A (Figure 3E).

Overall, our results show that endothelial progerin expression induced oxidative stress and was associated with accumulation of γ-H2AX foci, higher senescence-associated β-galactosidase activity and secreted proinflammatory cytokines, collectively known as senescence-associated secretory phenotype (SASP).

### 3.4. Inhibition of Progerin Prenylation Partly Prevents Endothelial Defects

Progerin permanent prenylation (farnesylation or alternative geranyl-geranylation) has been reported as a driver of HGPS disease progression and a potential pharmacological target [31,32]. We therefore hypothesized that blocking the synthesis of prenyl groups, using the combination of zoledronate and pravastatin (ZOPRA) [31], could prevent the endothelial cell alterations caused by progerin expression. In our model, ZOPRA treatment potently reduced the number of DNA damage-accumulating or senescent cells (Figure 4A,B). Inhibition of progerin prenylation using ZOPRA also reduced the proinflammatory/proadherent state of progerin-expressing cells as assessed by the reduction of the cellular secretion of IL6, CXCL8, CCL2, ICAM1 and VCAM1 (Figure 4C). All together these results demonstrate that progerin-induced endothelial cell senescence and inflammation is, at least in part, driven by the persistent prenylation of progerin.

## 4. Discussion

In this study we investigated the effect of progerin expression in human coronary endothelial cells issued from healthy donors. Our results demonstrate that progerin expression by itself triggers features of endothelial dysfunction that resemble those observed in the elderly population [17].

Our data are in favor of a direct role for progerin in the vascular wall of HGPS patients [10,12,17,33], independently of classical risk factors associated with cardiovascular diseases. Accordingly, patients with HGPS usually display only mildly altered glucose and lipid levels and blood pressure [3] with regard to the severity of their atherosclerosis lesions [3,9]. Of relevance, progerin is detectable at low levels in several tissues during normal aging, including atherosclerotic coronary arteries, and, therefore, could play a role in physiological aging [9,19]. In particular, progerin expression in aging endothelial cells could result in endothelial dysfunction and participate in the development of cardiovascular diseases within the elderly population.

Progerin overexpression in endothelial cells recapitulates the nuclear architecture defects observed in cells from patients with HGPS, notably characterized by nuclear blebs and invaginations [7]. In line with its permanent farnesylation, which increases its affinity for cell membranes, exogenous progerin was predominantly localized at the nuclear periphery.

Endothelial cell inflammation is an important contributor to atherosclerosis [34], notably during aging [35,36,37]. The presence of inflammation was also reported in the atherosclerotic lesions of patients with HGPS. Interestingly, mouse models of HGPS presented a hyperactivation of the proinflammatory transcription factor NF-kB (Nuclear Factor kappa-light-chain-enhancer of activated B cells) [33]. In addition, inhibition of the inflammatory response using *Sirt7* ectopic expression in endothelial cells significantly improved aging features in HGPS mice [38]. Herein, we show that endothelial progerin expression is sufficient to trigger a proinflammatory response with enhanced expression of the adhesion molecules that could participate in the recruitment of macrophage into the atherosclerotic plaque of HGPS patients [9], but also during physiological aging. Of relevance, endothelial oxidative stress is a known activator of NF-kB during aging [36] and reduces NO bioavailability in humans with coronary endothelial dysfunction [39]. In our model, endothelial progerin expression increases ROS production together with a proinflammatory response and a reduced endothelial nitric oxide synthase expression. Future studies may want to address the causative role of oxidative stress in the induction of inflammation and reduced NO synthase expression in progerin-expressing endothelial cells.

The cause of ROS accumulation induced by progerin remains to be elucidated in our model. As a potential mechanism, the nuclear lamina can serve as a ROS scavenger within the nucleus [40]. Oxidative stress in HGPS cells may also result from the reduced activity of the antioxidant transcription factor NRF2 [28], which is known to limit endothelial cell dysfunction and subsequent atherosclerosis [41,42,43]. Future work is required to fully understand the origin of ROS accumulation in endothelial cells expressing progerin.

In our study, we associated oxidative stress with the induction of DNA damage and cellular senescence, as previously observed in endothelial cells overexpressing mutated p.R482W lamin A or treated with protease inhibitor antiviral drugs, both resulting in the accumulation of farnesylated prelamin A [20,44]. These observations led us to investigate the ability of farnesylation blockade to prevent progerin-induced endothelial dysfunction. Although we have not assessed the efficiency of our ZOPRA dosage regarding inhibition of progerin prenylation, the dose used was previously shown to be efficient in reducing progerin prenylation [31]. In line with a role for the persistent farnesylation of progerin in inducing endothelial cells inflammation and senescence, the combination of the aminobisphosphonate zoledronate and the HMG-CoA reductase inhibitor pravastatin (ZOPRA) potently reduced the accumulation of DNA damage, cellular senescence, inflammation and the increased secretion of adhesion molecules caused by progerin overexpression in HCAECs.

Strategies to limit progerin farnesylation in HGPS cells used at first farnesyl transferase inhibitor (FTI). FTI successfully reduced nuclear blebbing in various cell models [23,45,46,47,48]. FTI also extends lifespan and reduces cardiovascular diseases in HGPS mouse models [49,50]. These promising results partly translated into clinic. Accordingly, treatment of HGPS patients with the FTI lonafarnib extended their lifespan [51,52] and notably reduced vascular stiffness [53]. However, alternative prenylation of the protein has been observed when FTI was used and could explain the moderate effect of FTI in HGPS patients [31]. The aforementioned study proposed, as an alternative strategy, to inhibit progerin prenylation by blocking the biosynthesis of both the farnesyl and geranyl residue with the combination of zoledronate and pravastatin (ZOPRA) [31]. Unfortunately, addition of ZOPRA to FTI only improved the bone mineral density of the patient, without further benefit regarding cardiovascular disease outcomes [54], suggesting a plateau effect of prenylation inhibition in HGPS patients. In the future, other therapies are considered, some of them in combination with prenylation inhibition, to improve the lifespan of HGPS patients [32].

Our study has limitations. Experiments were only performed in vitro and translation into clinic may be limited. Some of the results are normalized to the protein content per well and could not adequately reflect the number of senescent progerin-expressing cells. However, although senescence reduces the cellular proliferative capacity, ongoing protein synthesis in senescent cells could also result in an increased protein content par cell, therefore comforting our results expressed related to the protein content [55]. We have shown that, in our experiments, expression of progerin leads to a nonsignificant tendency to decrease the protein level per dish, so this should not modify the significance of our results. The level of progerin expression in our model was higher than that observed in progeria patients and in normal aging. However, overexpression of wild-type lamin A at a similar protein level had no adverse effects regarding endothelial cells function and senescence.

## 5. Conclusions

In conclusion, we propose that the progerin expression in endothelial cells of patients with HGPS could play a direct role in their premature atherogenesis. In addition, the presence of a low level of progerin in vascular cells during aging suggests that this protein could participate in physiological vascular aging, notably through induced endothelial dysfunction and senescence. In that setting, we propose that part of the cardioprotective effect of a treatment with a statin, often given to aging dyslipidemic patients, could also involve inhibition of progerin prenylation. Importantly, progerin-induced endothelial senescence, besides its cardiovascular consequences, could also triggers metabolic alterations in progeria mice models and reduces lifespan [18,38]. Endothelial progerin expression, therefore, is likely to be a determinant of aging-related dysfunction. Future studies may evaluate the potential of targeting progerin in the cardiometabolic-disease-prone elderly population.

## Figures and Tables

**Figure 1 cells-09-01201-f001:**
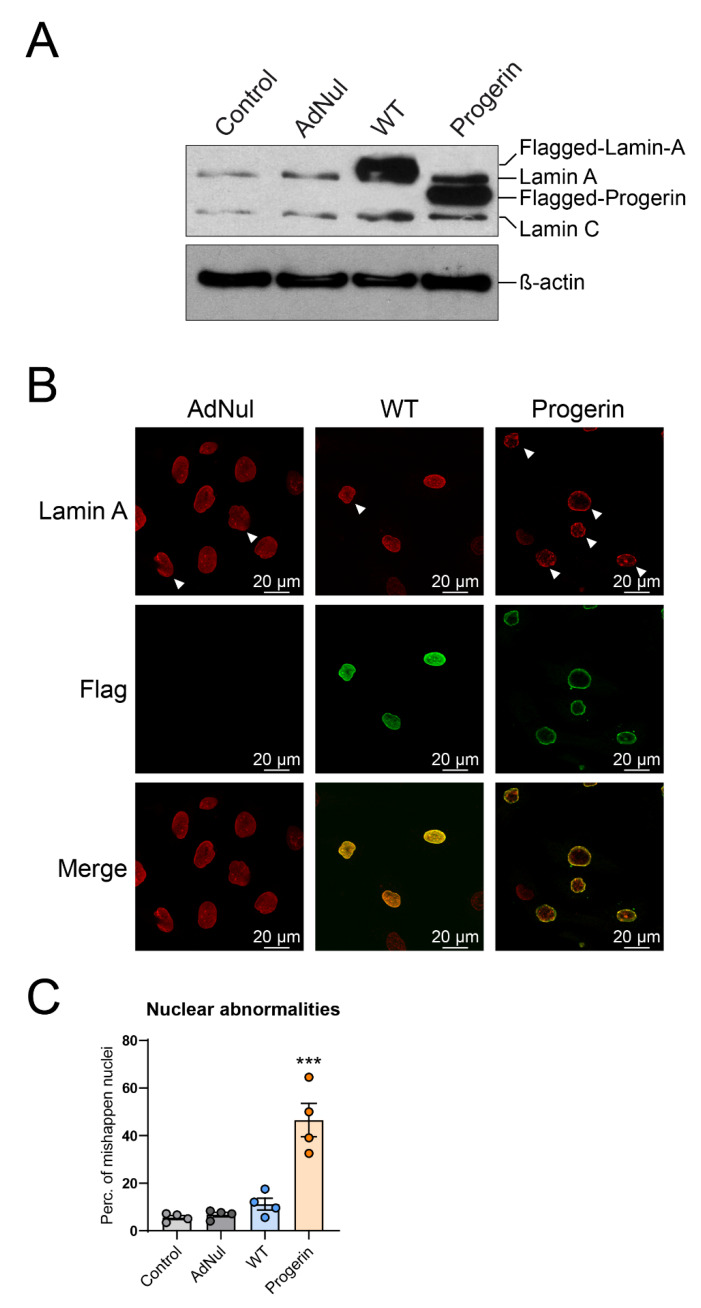
Expression and localization of exogenous lamin A and progerin. Early-confluent human coronary artery endothelial cells (HCAECs) were transduced or not with Flag-tagged recombinant adenovirus containing WT-prelamin A or progerin for 72 h, or with an empty vector (AdNul). (**A**) Western blot analysis of lamin A and C and progerin. β-actin was used as a loading control. Representative picture of *n* = 4 experiments. (**B**) Representative pictures of transduced HCAECs stained with Flag (green) and lamin-A (red) antibodies. Cells were observed by confocal microscopy at 100× magnification. Examples of misshapen nuclei are indicated with a white arrow. (**C**) Quantification of misshapen nuclei as percentage of total nuclei. Data are expressed as the mean ± standard error of mean (SEM) and statistical difference is determined using analysis of variance (ANOVA) followed by a Dunnett post hoc test. *** *p* < 0.001 vs. WT.

**Figure 2 cells-09-01201-f002:**
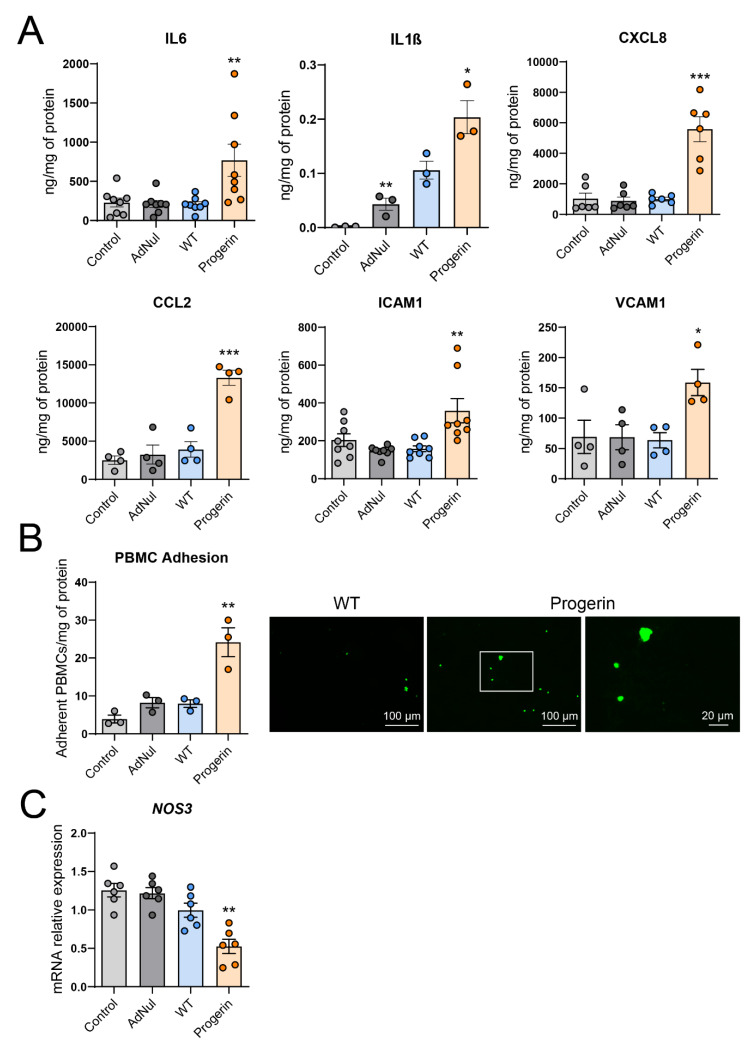
Progerin induces endothelial cells inflammation and dysfunction. Early-confluent HCAECs were transduced or not with Flag-tagged recombinant adenovirus containing WT-prelamin A or progerin for 72 h or with an empty vector. (**A**) Twenty-four hour secretion of the proinflammatory cytokines IL6, IL1β, of the chemokines CXCL8 and CCL2 and of the adhesion molecules ICAM1 and VCAM1. (**B**) Adhesion assay of peripheral blood mononuclear cells (PBMCs) from healthy donors on transduced and control endothelial cells was quantified as the number of adherent PBMC/mg of protein (left panel). Representative pictures are shown in the right panel with a magnification of the white squared area (**C**) Relative mRNA expression of the endothelial nitric oxide synthase gene (*NOS3*). Data are expressed as the mean ± SEM and statistical difference is determined using ANOVA followed by a Dunnett post hoc test. * *p* < 0.05, ** *p* < 0.01, *** *p* < 0.001 vs. WT.

**Figure 3 cells-09-01201-f003:**
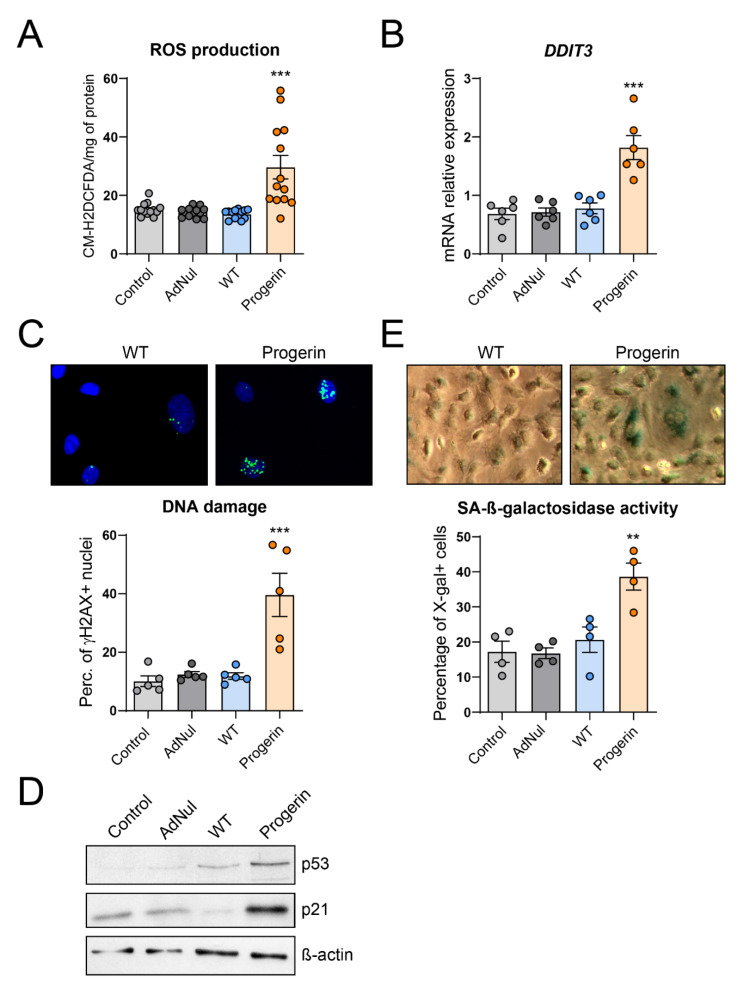
Endothelial progerin expression induces oxidative stress, DNA damage and cellular senescence. Early-confluent HCAECs were transduced or not with Flag-tagged recombinant adenovirus containing WT-prelamin A or progerin or with an empty vector. (**A**) Reactive oxygen species (ROS) production was assessed by the oxidation of 5-6-chloromethyl-2,7-dichlorodihydro-fluorescein diacetate (CM-H2DCFDA). (**B**) Relative mRNA expression of *DDIT3*. (**C**) DNA double-strand breaks (DSBs) were studied by staining HCAECs with Ser139-phosphorylated histone variant H2A (γ-H2AX, in green) and di-amidino-2-phenylindole hydrochloride (DAPI) (in blue) (upper panel) and evaluated as the percentage of γ-H2AX–positive cells (40–200 cells per experiment) (lower panel). (**D**) Protein expression of the cell cycle arrest proteins p53 and p21. Representative picture of *n* = 3 experiments. (**E**) Senescence-associated (SA)-β-galactosidase activity was assessed by the percentage of 5-bromo-4-chloro-3-indolyl-β-d-galactopyranoside (X-gal)-stained HCAECs (in blue) at pH6. Representative micrographs are shown (upper panel). Data are expressed as the mean ± SEM and statistical difference determined using ANOVA followed by a Dunnett post hoc test. ** *p* < 0.01, *** *p* < 0.001 vs. WT.

**Figure 4 cells-09-01201-f004:**
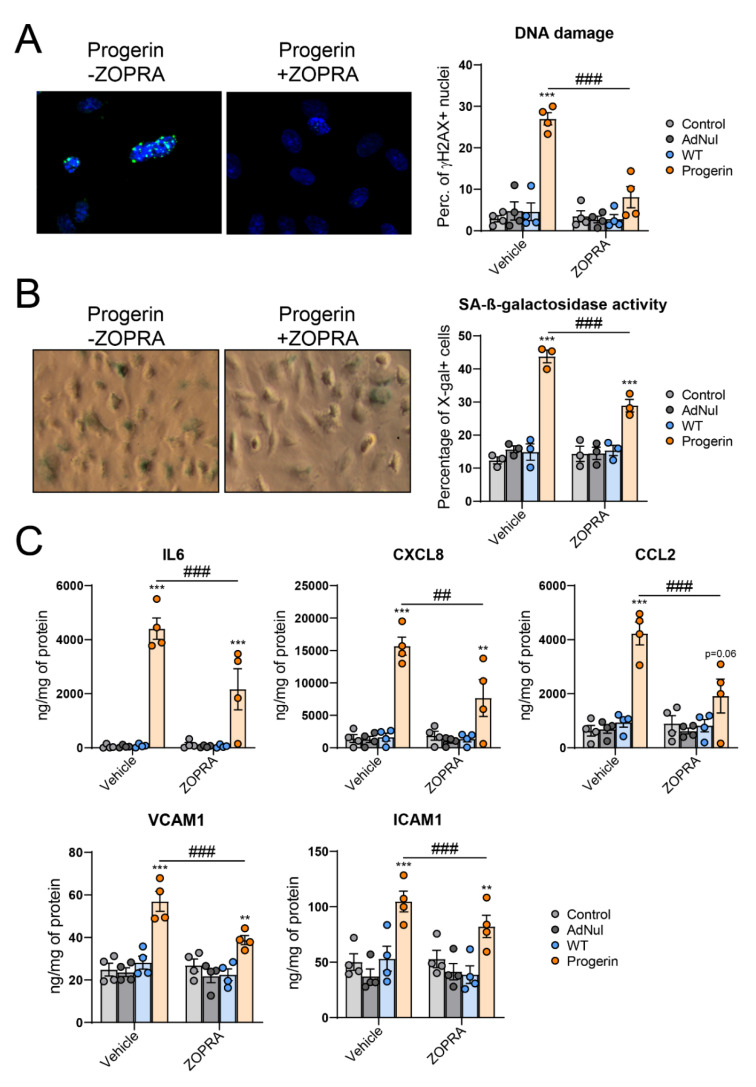
Inhibition of progerin prenylation partially prevents endothelial cells senescence, inflammation and secretion of adhesion molecules. Early-confluent HCAECs were transduced or not with Flag-tagged recombinant adenovirus containing WT-prelamin A or progerin for 72 h and immediately treated with zoledronate and pravastatin (ZOPRA). (**A**) DNA DSBs were studied by staining HCAECs with γ-H2AX (in green) and DAPI (in blue). Representative micrographs of HCAECs expressing progerin are shown (left panel). DNA DSBs were quantified as the percentage of γ-H2AX–positive cells (40–200 cells per experiment) (right panel). (**B**) Senescence-associated (SA)-β-galactosidase activity was assessed as the percentage of X-gal–stained cells (blue) at pH6 (right panel). Representative micrographs are shown (left panel). (**C**) 24 h-secretion of the proinflammatory cytokine IL6, of the chemokines CXCL8 and CCL2 and of the adhesion molecules ICAM1 and VCAM1. Data are expressed as the mean ± SEM and statistical difference was determined using two-way ANOVA followed by a Tukey post hoc test to assess differences against WT-prelamin A overexpression or a Sidak post hoc test to determine ZOPRA effect for each condition. *** *p* < 0.001 vs. WT. ## *p* < 0.01, ### *p* < 0.001 vs. vehicle-treated cells.

**Table 1 cells-09-01201-t001:** Primer list.

Gene	Forward (5′ to 3′)	Reverse (5′ to 3′)
*HPRT*	TAATTGGTGGAGATGATCTCTCAAC	TGCCTGACCAAGGAAAGC
*IL6*	CACACAGACAGCCACTCACC	CATCCATCTTTTTCAGCCATC
*IL1b*	TACCTGTCCTGCGTGTTGAA	TCTTTGGGTAATTTTTGGGATCT
*CXCL8*	AGACAGCAGAGCACACAAGC	ATGGTTCCTTCCGGTGGT
*CCL2*	TCAGCCAGATGCAATCAATG	TCCTGAACCCACTTCTGCTT
*ICAM1*	TGGTAGCAGCCGCAGTCATA	CTCCTTCCTCTTGGCTTAGT
*VCAM1*	5CGCTGACAATGAATCCTGTTAGT	GTATTCTTGGGTGATATGTAGACTTG
*DDIT3*	AAACGGAAACAGAGTGGTCATTCCCC	GTGGGATTGAGGGTCACATCATTGGCA
*NOS3*	GTGGCTGTCTGCATGGACCT	CCACGATGGTGACTTTGGCT

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
