# Peer review of "Progerin Expression Induces Inflammation, Oxidative Stress and Senescence in Human Coronary Endothelial Cells"

_cells, 2020, doi:10.3390/cells9051201_

Round 1

Reviewer 1 Report

Animal models have established the importance of endothelial cells in the initiation of the fatty streak lesion and progression of atherosclerosis development. In this study, the authors examined the effects of progerin (the abnormal nuclear lamin protein produced in progeria) on endothelial cell function. The effects of progerin were examined in transiently transduced HAECs, and compared to the effects of wild type (WT) prelamin A, which is processed to mature lamin A.

  1. Although the average transduction efficiency for progerin and WT prelamin A was very similar (Suppl. Fig. 1), the range for both was large (10-70% and 30-80%, respectively). Thus, the levels of expression for progerin and prelamin A could be very different in individual experiments. How did the authors ensure the levels of expression for prelamin A and progerin were matched in the different studies? Were western blots performed, or other?
  2. For multiple studies the results are expressed relative to endothelial cell protein (i.e., Figs. 2A, 2B, 3A, 4C). As the authors correctly point out, progerin causes DNA damage, cell cycle arrest, and cell senescence. This raises the possibility that cell numbers (and the amount of cell protein) could be lower in the progerin group. This could amplify the quantitative effects of progerin (e.g., higher cytokine production or higher adhesion molecule secretion). Were the levels of cell protein affected by progerin expression? A statement should be added to the Methods or Results clarifying whether progerin expression affected cell protein levels.
  3. The adhesion of PBMCs to endothelial cells was assessed by fluorescence microscopy. Based on the microscopy image in Fig. 2B, it is clear that there are more green spots in the progerin group. However, it is difficult to determine if the green spots are actually cells. Is it possible to improve the quality of the image (e.g., adding an inset at higher magnification)? At a minimum, a scale bar should be added. The authors should also add a comment in the Results regarding the identity of the adherent cells; are they monocytes, lymphocytes, or undetermined.
  4. The authors report that the secretion of VCAM1 and ICAM1 into the culture supernatant was increased in progerin-expressing cells (Fig. 4C). Did the authors measure adhesion molecule expression on the cell surface? If not, the authors should comment whether levels of soluble VCAM1/ICAM parallel their expression on the cell surface.
  5. The improvement in endothelial cell function with ZOPRA is consistent with the proposal that the prenylation of progerin contributes to its toxicity. Do the authors have any evidence that the treatment actually affected protein prenylation (e.g., Verstraeten VL et al, PNAS 108:4997)?

Reviewer 2 Report

This study provides new data regarding vascular dysfunction in progeria. In particular, Bidault et al propose that progerin expression in human endothelial cells can recapitulate features of endothelial dysfunction, which is known to be determinant in vascular disease promoting atherosclerosis during aging. The authors have illustrated the effect of progerin expression in human coronary endothelial  cells (HCAECs) triggering inflammation, DNA damage, and senescence, ultimately driving endothelial dysfunction. They also have demonstrated how a progerin farnesylation inhibitor (ZOPRA)  partially prevents some progerin-induced endothelial cells alterations.

Data showing the effects of progerin expression in human endothelial cells are sound and novel.

The manuscript is well-written and  the experiments well-controlled. Although for the most part, the data support the claims of the study, a weakness is identified in this work regarding how progerin is directly impacting endothelial dysfunction. It is known that sustained inflammation associated with enhanced oxidative stress is likely to be a major cause for endothelial dysfunction. In the context of vascular disease, endothelial dysfunction usually is characterize by reduced dilatory capacities caused by reduced nitric oxide (NO) levels, the most important vasodilator molecule. The authors have analyzed expression of a key protein producing NO, the nitric oxide synthase (NO3), upon progerin expression. They showed a significantly decrease in NO3 levels upon progerin expression, relating progerin expression with  endothelial dysfunction. However, data showing whether inhibiting progerin, endothelial dysfunction is rescued would strengthen the conclusions and the overall impact of this manuscript.

Addressing other aspects would also improve the manuscript:

  1. In figure 1, Bidault et al transiently transduced HCAECs with adenoviral vector containing the cDNA of flag-tagged WT-prelamin A or Progerin. Progerin was accumulated specifically at the nuclear rim, causing nuclear abnormalities as previously observed in other models. Some quantification of nuclear abnormalities (i.e. blebs, invaginations, or nuclear size) would strengthen the results.

  1. In figure 2, the authors have successfully demonstrated that progerin expression induces endothelial cell inflammation and dysfunction: they have shown an increase in different inflammatory markers and a decrease in NOS3 expression. Moreover,  progerin-expressing HCAEC cells exhibit increased PBMC adhesion. The quantification of the number of PBMC recruited was normalized using mg of protein. I suggest presenting a more accurate image of the number of cells (i.e. DAPI stained cells), giving  some idea of PBMC recruited/number of cells.

  1. Combination of zoledronate and pravastatin (ZOPRA) partially rescue some endothelial cell alterations caused by progerin expression such as DNA damage, senescence and inflammation. However, prevention at least in part of endothelial dysfunction (NO3 expression) or oxidative stress has not been tested. There is also a significant decrease in inflammatory cytokines,  ICAM and VCAM expression in ZOPRA treated cells. Whether or not this translates into changes in PBMC adhesion is not tested either. These experiments would improve the significance of the study.

  1. Minor comments: add magnification in the pictures

Author Response

REVIEWER 2

Comments and Suggestions for Authors

This study provides new data regarding vascular dysfunction in progeria. In particular, Bidault et al propose that progerin expression in human endothelial cells can recapitulate features of endothelial dysfunction, which is known to be determinant in vascular disease promoting atherosclerosis during aging. The authors have illustrated the effect of progerin expression in human coronary endothelial  cells (HCAECs) triggering inflammation, DNA damage, and senescence, ultimately driving endothelial dysfunction. They also have demonstrated how a progerin farnesylation inhibitor (ZOPRA)  partially prevents some progerin-induced endothelial cells alterations.

Data showing the effects of progerin expression in human endothelial cells are sound and novel. The manuscript is well-written and  the experiments well-controlled. Although for the most part, the data support the claims of the study, a weakness is identified in this work regarding how progerin is directly impacting endothelial dysfunction. It is known that sustained inflammation associated with enhanced oxidative stress is likely to be a major cause for endothelial dysfunction. In the context of vascular disease, endothelial dysfunction usually is characterize by reduced dilatory capacities caused by reduced nitric oxide (NO) levels, the most important vasodilator molecule. The authors have analyzed expression of a key protein producing NO, the nitric oxide synthase (NO3), upon progerin expression. They showed a significantly decrease in NO3 levels upon progerin expression, relating progerin expression with  endothelial dysfunction. However, data showing whether inhibiting progerin, endothelial dysfunction is rescued would strengthen the conclusions and the overall impact of this manuscript.

We thank the reviewer for her/his detailed assessment of our manuscript and positive comments.

 Addressing other aspects would also improve the manuscript:

Point 1 In figure 1, Bidault et al transiently transduced HCAECs with adenoviral vector containing the cDNA of flag-tagged WT-prelamin A or Progerin. Progerin was accumulated specifically at the nuclear rim, causing nuclear abnormalities as previously observed in other models. Some quantification of nuclear abnormalities (i.e. blebs, invaginations, or nuclear size) would strengthen the results.

Response 1: We have added in the revised version of the manuscript the percentage of nuclear abnormalities (Fig. 1C and Material and methods section (Line 117-118)).

Point 2 In figure 2, the authors have successfully demonstrated that progerin expression induces endothelial cell inflammation and dysfunction: they have shown an increase in different inflammatory markers and a decrease in NOS3 expression. Moreover,  progerin-expressing HCAEC cells exhibit increased PBMC adhesion. The quantification of the number of PBMC recruited was normalized using mg of protein. I suggest presenting a more accurate image of the number of cells (i.e. DAPI stained cells), giving  some idea of PBMC recruited/number of cells.

Response 2: We acknowledge that a quantification of the number of cells would be interesting. However, we deliberately decided to normalise our data by protein content, to avoid superposition in the DAPI channel between adherent PBMC and adherent cells.

Point 3 Combination of zoledronate and pravastatin (ZOPRA) partially rescue some endothelial cell alterations caused by progerin expression such as DNA damage, senescence and inflammation. However, prevention at least in part of endothelial dysfunction (NO3 expression) or oxidative stress has not been tested. There is also a significant decrease in inflammatory cytokines,  ICAM and VCAM expression in ZOPRA treated cells. Whether or not this translates into changes in PBMC adhesion is not tested either. These experiments would improve the significance of the study.

Response 3: We agree with the reviewer that assessing the adhesion of PBMC in ZOPRA-treated cells would improve the value of our study. However, we show in Figure 2 a good association between cell adhesion and the endothelial activation induced by progerin and therefore we believe that a reduction of adhesion to PBMC would be observed in ZOPRA-treated cells. Unfortunately, due to COVID, our laboratory is closed and we cannot perform any experiment.

Minor comments: add magnification in the pictures

Response Minor comments: Scale bars have been added to confocal microscopy pictures (Fig. 1B). In addition, as required by reviewer 1, we added magnified pictures of the cell layer indicating that the coloured cells have a size compatible with that of PBMC.

Round 2

Reviewer 1 Report

Thank you for responding to my concerns.